# Cost Function Analysis Applied to Different Kinetic Release Models of *Arrabidaea chica* Verlot Extract from Chitosan/Alginate Membranes

**DOI:** 10.3390/polym14061109

**Published:** 2022-03-10

**Authors:** Luis Concha, Ana Luiza Resende Pires, Angela Maria Moraes, Elizabeth Mas-Hernández, Stefan Berres, Jacobo Hernandez-Montelongo

**Affiliations:** 1Department of Physical and Mathematical Sciences, Catholic University of Temuco, Temuco 4813302, Chile; luisconcha.c@gmail.com; 2School of Chemical Engineering, University of Campinas, Campinas 13083-852, Brazil; analurespi@gmail.com (A.L.R.P.); ammoraes@unicamp.br (A.M.M.); 3Department of Mathematical Engineering, University of La Frontera, Temuco 4811230, Chile; elimher@gmail.com; 4Bioproducts and Advanced Materials Research Nucleus (BioMA), Catholic University of Temuco, Temuco 4813302, Chile; 5Department of Information Systems, University of Bio-Bio, Concepcion 4051381, Chile

**Keywords:** cost function, controlled release, *Arrabidae chica* Verlot, chitosan/alginate membranes

## Abstract

This work focuses on the mathematical analysis of the controlled release of a standardized extract of *A. chica* from chitosan/alginate (C/A) membranes, which can be used for the treatment of skin lesions. Four different types of C/A membranes were tested: a dense membrane (CA), a dense and flexible membrane (CAS), a porous membrane (CAP) and a porous and flexible membrane (CAPS). The *Arrabidae chica* extract release profiles were obtained experimentally in vitro using PBS at 37 °C and pH 7. Experimental data of release kinetics were analyzed using five classical models from the literature: Zero Order, First Order, Higuchi, Korsmeyer–Peppas and Weibull functions. Results for the Korsmeyer–Peppas model showed that the release of *A. chica* extract from four membrane formulations was by a diffusion through a partially swollen matrix and through a water filled network mesh; however, the Weibull model suggested that non-porous membranes (CA and CAS) had fractal geometry and that porous membranes (CAP and CAPS) have highly disorganized structures. Nevertheless, by applying an explicit optimization method that employs a cost function to determine the model parameters that best fit to experimental data, the results indicated that the Weibull model showed the best simulation for the release profiles from the four membranes: CA, CAS and CAP presented Fickian diffusion through a polymeric matrix of fractal geometry, and only the CAPS membrane showed a highly disordered matrix. The use of this cost function optimization had the significant advantage of higher fitting sensitivity.

## 1. Introduction

*Arrabidaea chica* Verlot is a type of shrub found in tropical America, from the south of Mexico to Brazil, being very common in the Amazon rainforest [1]. As the extract of *A. chica* is an important source of tannins, flavonoids and anthocyanins, it presents different medicinal properties, such as antioxidant, antiseptic, anti-inflammatory and antifungal activities [2]. One of its main components is the anthocyanin ‘carajurina’, which can be used as a marker for the detection and quantification of the extract of *A. chica*.

One of the strategies to deliver drugs and other medicinal substances is through controlled release systems, which maintain the drug concentration in the blood or in target tissues as long as possible at a desired value, being able to control the drug release rate and duration [3]. Different biopolymers have been used to deliver drugs and other compounds for the treatment of skin lesions in a local, targeted and controlled manner [4]. One of the most versatile biopolymers is chitosan, which is obtained mainly from shells of crustaceans, such as shrimp [5]. In this regard, Servat-Medina et al. (2015) [6] synthesized chitosan nanoparticles loaded with different concentrations of *A. chica* extract (10 to 25% relative to chitosan mass) for the treatment of skin ulcers. Recently, in 2020, our group reported synthesizing different types of chitosan/alginate (C/A) membranes to be used as controlled release systems of *A. chica* extract, as an alternative for the treatment of skin lesions [7]. C/A membranes are attractive because they are insoluble in water, stable to pH variations and capable of incorporating different bioactive agents into their matrix.

Mathematical modeling of drug delivery and predictability of drug release has been a field of academic and industrial importance for several decades [8]. In 1961, Higuchi published his famous equation allowing for a surprisingly simple description of the drug release mechanism from an ointment base. Numerous models have been proposed since then, including empirical/semi-empirical as well as mechanistic realistic ones [9]. Consequently, many different mathematical approaches have been proposed to assess the similarity between drug dissolution and mass transfer profiles [10].

A versatile mathematical tool is the cost function analysis, which is an optimization process that consists of measuring the difference between real experimental data with the prediction of a model. The objective of this analysis is to minimize this difference, that is, to find the parameters that allow the model to fit the data as closely as possible [11]. The cost function can be performed for any type of mathematical model and compared with various experimental data; regarding numerical optimization methods, local gradient methods such as conjugate gradients [12,13] or global parameter population methods such as genetic algorithms [11,14] or deep learning strategies [15] can be applied.

The method in this work focuses on the quantitative assessment of the controlled release of a standardized extract of *A. chica* from C/A membranes. First, in vitro experimental *A. chica* kinetic release profiles were obtained from four different formulations of C/A membranes: a dense membrane (CA), a dense and flexible membrane (CAS), a porous membrane (CAP), and a porous and flexible membrane (CAPS). Later, the mechanisms of the extract release kinetics were determined comparing the $r^{2}$ coefficient obtained using five classic models from the literature [16]: Zero Order, First Order, Higuchi, Korsmeyer–Peppas and Weibull model functions. Finally, as the main novelty of this work, the release profiles were analyzed using an optimization method implemented by our group that employs the cost function approach to determine the model parameters that best fit to the experimental data.

## 2. Materials and Methods

### 2.1. Materials

Chitosan (C, from shrimp shells 96% deacetylated and $M_{W}=1.26\times 10^{6}$ g/mol), medium viscosity sodium alginate (A, from *Macrocystis pyrifera* $M_{W}=9.11\times 10^{4}$ g/mol), Kolliphor P188 (a pore-forming surfactant) and phosphate buffer solution (PBS) 0.01 M (0.138 M NaCl, 0.0027 M KCl, pH = 7.4) were acquired from Sigma-Aldrich (Sao Paulo, Brazil). Silpuran 2130 A/B (a silicone polymer) was obtained from Wacker Chemie AG (Munich, Germany), and glacial acetic acid, calcium chloride dihydrate and sodium salt from Merck KGaA (Sao Paulo, Brazil). The standardized *Arrabidaea chica* Verlot extract was supplied by the Division of Chemistry of Natural Products of the Center for Chemical, Biological and Agricultural Research (CPQBA) at the University of Campinas (Campinas, Brazil). The used water was deionized in a Milli-Q System from Millipore.

### 2.2. Membranes synthesis

Four formulations of C/A membranes containing *A. chica* extract (10% in weight) were obtained according to the protocol previously described by Pires et al. (2020) [7]. The main synthesis differences are summarized as follows: (A) CA membrane: prepared with chitosan 1% (*m/v*) in acetic acid 2% and alginate 0.5% (C:A = 1:2 *v/v*); (B) CAS membrane: synthesized with the same CA formulation, but including 10% of Silpuran 2130 A/B; (C) CAP membrane: obtained by formulating CA membrane plus 10% in weight of the Kolliphor P188; (D) CAPS membrane: It was synthesized by formulating CAS plus 10% in weight of Kolliphor P188. In all cases, the membranes were cross-linked with calcium ions.

### 2.3. Morphology of the Membranes

Membrane samples containing the *A. chica* extract were microscopically observed and photographed using a Nikon digital camera (COOLPIX model S3300). The cross section morphology of the membranes was analyzed using a scanning electron microscope (model Leo 440i, Leica). Samples of 2cm×1cm were fixed on a suitable support and metalized (mini-Sputter coater, SC 7620) by depositing a thin layer of gold (92 Å) on their surfaces.

### 2.4. Release Experiments

To obtain the release profiles, membrane samples (2 cm × 2 cm) containing the *A. chica* extract were previously weighed and immersed in a 10 mL of PBS solution containing 20% ethanol at pH 7.4, $37{\;}^{\circ }$C and 100 rpm. At predetermined time intervals up to 48 h, 1 mL of the solution was withdrawn for absorbance analysis using a spectrophotometer (Thermo Scientific Evolution–220, Thermo Fisher Scientific, Waltham, MA, USA) at 470 nm and then returned to the vial. The analytical curve was prepared using *A. chica* extract also dissolved in a PBS solution containing 20% ethanol. All experiments were performed in triplicate, and mean values were used. Sink conditions were maintained during the drug release experiments and at predetermined time intervals, the supernatant solution was completely renewed.

### 2.5. Mathematical Models

The mechanism of drug release was determined by fitting the mathematical models to the experimental data using OriginPro 8.5 software. Five models were studied, the main characteristics of which are outlined below [17,18,19,20].

In the zero-order model, the drug is released at a constant rate independent of concentration, and dissolution from dosage forms that do not disaggregate and release the drug slowly. This model is represented by the equation
(1)$$\begin{array}{c} Q\left(t\right)=Q_{0}+k_{0}t, \end{array}$$
where Q(t) is the amount of drug released at time *t*, $Q_{0}=Q\left(t=0\right)$ is the initial amount of the extract in the solution and $k_{0}$ is the zero-order proportional constant.

In the first-order model, the release is a concentration-dependent process, and the equation that gives the release behavior is
(2)$$\begin{array}{c} Q\left(t\right)=Q_{0}\left(1-\exp\left(-k_{1}t\right)\right), \end{array}$$
where Q(t) and $Q_{0}$ are again the amount of drug released at time *t* and the initial amount of the extract in the solution, respectively, and $k_{1}$ is the first-order release kinetic constant.

The Higuchi model implies more assumptions, such as that the initial drug concentration in the matrix is higher than drug solubility; drug diffusion takes place only in one dimension (edge effect must be negligible); drug particles are smaller than system thickness; matrix swelling and dissolution are negligible; drug diffusivity is constant; and perfect sink conditions are always attained in the release environment. The general release equation is given by
(3)$$\begin{array}{c} Q\left(t\right)=k_{H}t^{1/2}, \end{array}$$
where Q(t) is the amount of drug released in time *t* per unit area, and $k_{H}$ is the Higuchi dissolution constant. It represents a Fickian diffusion of drugs without the matrix dissolution taken into account.

The Korsmeyer–Peppas model is a simple relationship to describe drug release from a polymeric system. This semi-empirical model analyzes both Fickian and non-Fickian release of drug from swelling as well as non-swelling materials; however, it is applied just up to 60% of the drug amount released. The Korsmeyer–Peppas release equation is given by
(4)$$\begin{array}{c} \frac{Q(t)}{Q_{\infty }}=k_{\mathrm{KP}}t^{n}, \end{array}$$
where the ratio $Q\left(t\right)/Q_{\infty }$ is the fraction of drug released at time *t*, $k_{\mathrm{KP}}$ is the Korsmeyer–Peppas kinetic constant, which characterizes the drug–matrix system and *n* is the exponent that indicates the drug release mechanism.

The Weibull model is an alternative description for the dissolution and release processes, and can be applied for most types of dissolution curves. The Weibull equation expresses the cumulative fraction of the drug, Q(t), in a solution at time *t*, by the following expression:(5)$$\begin{array}{c} Q\left(t\right)=1-\exp\left(-{\left(t-T_{i}\right)}^{\beta }/\alpha \right), \end{array}$$
where α is related to the specific surface of the dosage matrix form, β is mainly related to the mass transport characteristics of the device and $T_{i}$ represents the delay time before starting the dissolution or release process, which in most cases is 0.

## 3. Cost function Analysis

### 3.1. Definition of Cost Function

We introduce the notation Q(e;t) for a general model that depends on the parameter vector e. For example, for the Korsmeyer–Peppas model, the parameter set is specified as
$$\begin{array}{c} e=\left(e_{1},e_{2}\right)=\left(k_{\mathrm{KP}},n\right). \end{array}$$

A standard technique that enables the interpretation of experimental data with a quantitative model is the formulation as an inverse problem, where the direct problem is formulated as a mathematical model, such that the distance of the model $Q(e;t_{i})$ to the data
$$\begin{array}{c} \left(t_{i},{\hat{Q}}_{i}\right),\;i=1,\cdots ,N \end{array}$$
is described by a cost function [11,21]
(6)$$\begin{array}{c} F\left(e\right)=\sum_{i=1}^{n}\mu_{i}{\left|Q\left(e;t_{i}\right)-{\hat{Q}}_{i}\right|}^{p}, \end{array}$$
where p∈{1,2,⋯} accounts for different metrics of the distance between model and data, and $\mu_{i}=\mu \left(t_{i}\right)$ contains weights of the data points. For p=2 (in comparison to p=1), we have an underestimation respective overestimation of the measurement errors (under the assumption that the model is correct) for small respective high distances between data and model. Though, p=2 applies the concept of least squares, which is more common, in spite of the bias.

Regarding the weights, there are various choices, as the following prototypes, among which there are multiple possible variants:Equal weights $\mu_{i}\equiv 1$ for all *i*.Switch off at a threshold time $t^{*}$,
$$\begin{array}{c} \mu_{i}=\left\{\begin{array}{cc} 1 & \;\mathrm{for}\;t_{i}<t^{*}\\ 0 & \;\mathrm{for}\;t_{i}>t^{*}, \end{array}\right. \end{array}$$Adaptive weights, that gives higher emphasis to smaller times, such as, for example,
$$\begin{array}{c} \mu \left(t_{i}\right)=\frac{1}{c+t_{i}},\;c>0. \end{array}$$

The switch off in case (2) at a threshold time $t^{*}$ applies, i.e., should apply, for model approximations that do not satisfy asymptotic upper limits, such as the models of Higuchi or Korsmeyer–Peppas, which are designed to be valid for lower times only. As their unlimited asymptotic behavior is qualitatively wrong, above a threshold value the models are quantitatively wrong. In favor of the Higuchi and Korsmeyer–Peppas models, the weighting choice was case (2) with data switch off at $t^{*}=8$ h. For the Weibull model we choose case (1).

The inverse problem is solved by minimizing the cost function,
(7)$$\begin{array}{c} \min_{e\in R^{n}}F\left(e\right), \end{array}$$
that gives the optimal set of parameters e. The mathematical model can have the form of differential equations that in turn include parametric functions, or the mathematical model is expressed directly in terms of parametric functions. For the minimization of the cost function, an issue might be (1) high correlations between the parameters, or (2) its non-convexity. In the case of high parameter correlation that touch the scale of computational errors induced by the machine error, even the optimal numerical method cannot compensate a wrong model choice; it is an issue of model choice. In the case of non-convexity of the cost function, the optimization method needs to be chosen carefully, or the experimentation with different optimization methods is subject of research, e.g., global optimization methods, where populations of parameter sets are optimized, instead of local methods with only one single parameter set.

### 3.2. Cost Function Minimization: Optimality Conditions

The goal is to minimize the cost function, that is, to find the parameters of each model that fits the experimental data as closely as possible.

As a criterion of optimality [22], there is a necessary condition and a sufficient condition. For two-variable models, the necessary condition for optimality is that the gradient is equal to zero in each of its components,
(8)$$\nabla f\left(e_{1},e_{2}\right)=\left(\begin{array}{c} \frac{\partial }{\partial e_{1}}f\left(e_{1},e_{2}\right)\\ \frac{\partial }{\partial e_{2}}f\left(e_{1},e_{2}\right) \end{array}\right)=\left(\begin{array}{c} 0\\ 0 \end{array}\right).$$

The sufficient condition of optimality is that the Hessian matrix
(9)$$\begin{array}{c} H\left(e_{1},e_{2}\right)=\left(\begin{array}{cc} \partial_{e_{1}e_{1}}f\left(e_{1},e_{1}\right) & \partial_{e_{1}e_{2}}f\left(e_{1},e_{2}\right)\\ \partial_{e_{2}e_{1}}f\left(e_{1},e_{2}\right) & \partial_{e_{2}e_{2}}f\left(e_{2},e_{2}\right) \end{array}\right) \end{array}$$
is positive definite for a minimum, i.e., according to the definition
(10)$$\left(\begin{array}{cc} q_{1} & q_{2} \end{array}\right)H\left(e_{1},e_{2}\right)\left(\begin{array}{c} q_{1}\\ q_{2} \end{array}\right)>0,\;\forall q_{1},q_{2}\in R^{2}.$$

This definition can be written as
(11)$$\begin{array}{c} q_{1}^{2}f_{e_{1}e_{1}}+2q_{1}q_{2}f_{e_{1}e_{2}}+q_{2}^{2}f_{e_{2}e_{2}}>0,\;\forall q_{1},q_{2}. \end{array}$$

A general criterion of a square matrix of any size to be positive definite is that all eigenvalues are positive.

### 3.3. Equivalence of Optimization Methods

The comparison of model predictions with data,
(12)$$\begin{array}{c} f_{i}\left(e\right)=Q\left(e,t_{i}\right)-{\hat{Q}}_{i},\;i=1,\cdots ,N, \end{array}$$
defines a residual function $f:R^{n}\to R^{N}$, where we want to find its zeros, which can only be calculated approximately if the system is overdetermined. An approximation criterion is the sum of squares
(13)$$\begin{array}{c} F\left(e\right)=\sum_{i=1}^{N}f_{i}^{2}\left(e\right), \end{array}$$
which is equivalent to the cost function (Equation 6), but for the specific choice of p=2 and with equal weights, i.e. case (1).

There are two different paths to obtain optimal parameters:Approximate the zeros of the overdetermined system (Equation 12).Calculate the zeros of the gradient of the cost function (Equation 13), defined as the sum of squares of the residual.

An iterative procedure that solves an overdetermined system of nonlinear equations, such as (Equation 6) is the Gauss–Newton algorithm. This means that an initial estimate of the vector parameter must be provided.

## 4. Methodology: Implementation

The five mathematical models of Section 2.5 were tested in parallel. The generated process was automated by subroutines (see Appendix A).

Prior to the calculations, the derivatives and the gradients were obtained for one- and two-variable models, respectively, in order to verify the fulfillment of the necessary condition. The sufficient condition was satisfied, as for one-variable models the second derivative, and for two-variable models the elements in the Hessian matrix were different from zero.

In the implementation, the corresponding equations and the experimental data were consistent enough to be called by the common cost function subroutine.

The algorithm for the cost function is given in the Appendix A (see Appendix A for Algorithms). The following criterion has to be satisfied:(14)$$\begin{array}{c} c=\sum_{i=1}^{N}\sum_{j=1}^{J}{\left|u_{i}\left(t_{j}\right)-{\hat{u}}_{ij}\right|}^{2}, \end{array}$$
where *c* is the cost function, *N* is the number of experiments for each membrane in a given time *t* (in our case N=3) and *J* is the number of data in the experiment. A syntax was chosen from the model subroutine (Appendix A):$$u=(\mathrm{uModel}\;\hat{t}\;),$$
where the vector of variables consisted of the observation times, to calculate the solution of the model at given times. The description of steps to implement this methodology is presented in Appendix A.

It is important to consider that a better fit is expected with the cost function used in models with several parameters. However, that does not necessarily mean that the type of model as such is better or that it effectively better explains the phenomenon. Moreover, a higher number of parameters causes optimization algorithms to have problems when approaching the ‘optimal’ data, given the higher correlation within the same parameters.

## 5. Results and Discussion

The obtained membranes presented the geometrical form of a thin slab. Figure 1 shows photographs and SEM images of the membrane samples loaded with *A. chica* extract. The observed intense red color in the photographs (Figure 1A–D) was due to the insoluble anthocyanin pigments carajurin and carajurone included in the *A. chica* extract [23]. The membranes produced without Kolliphor P188 surfactant turned out to be dense and compact (CA and CAS). However, for the formulations synthesized using the surfactant, the membranes were porous and thicker (CAP and CAPS). When the silicone Silpuran 2130 was included in the formulation, samples were soft and flexible to the touch (CAS and CAPS). In summary, the general characteristics of the obtained membranes were as follows: CA was dense, thin and rigid; CAS was dense, thin but flexible due to the silicone; CAP was rigid but thick and porous due to air bubbles in its matrix generated by the surfactant; and CAPS was thick and porous due to the surfactant, but flexible because of the silicone.

*A. chica* extract release profiles are shown in Figure 2. Although all samples were loaded with the same percentage of extract (10% in weight), CA and CAS membranes released higher amounts of extract per mass of polymer than CAP and CAPS. The burst release observed in the first minutes of the process can be attributed to the fraction of the drug which is adsorbed or weakly bound to the surface area of the polymer rather than to the drug incorporated into the polymer matrix [24]. Moreover, the lower amounts in porous formulations can be explained by the higher dispersion of the mixture due to the presence of the surfactant with a consequent increase in the interphases of the extract with the polymers. For all cases, the maximum amount of extract released was reached at 24 h. Re-absorption of the extract by the membranes was not observed because the maximum released values of extract were maintained constant for up to 48 h.

To gain a deeper insight into the mechanisms that govern the release of *A. chica* extract from the membranes, five mathematical models were fitted to the experimental data: Zero-order, First order, Higuchi, Korsmeyer–Peppas and Weibull. Experimental data were normalized and only evaluated up to 8 h because these models are semi-empiric and better valid for the first stages of release; as mentioned previously in Section 3.1, where the weights on the cost function switch off (Equation 7) at a threshold value of time $t^{*}$.

The switch off is an artificial fix for a model that is valid for short time intervals and loses pertinence at bigger times. In order to avoid this artificial switch off, one can select admissible functions for a semi-empirical model that satisfy physically reasonable criteria. For the drug release model, these characteristics should address at least the following components:Zero release at zero time.Increasing cumulative release for advancing time.Existence of an upper bound for the total release at sufficiently long period.

These characteristics can be expressed as
$$\begin{array}{c} Q\left(0\right)=0,\;\frac{\partial Q(t)}{\partial t}>0,\;\lim_{t\to \infty }Q\left(t\right)<\infty . \end{array}$$

With help of these model design indications, we can assess the various models (see Table 1 for a qualitative model comparison).

Simulations of the release profiles from each membrane using the indicated models are shown in Figure 3. From these simulations, the kinetic parameters were extracted and presented in Table 2. The $r^{2}$ coefficient (coefficient of determination) was used to compare the results between models. The value of $r^{2}$ is usually between 0 and 1, and generally a higher value means that the model fits the data better; some $r^{2}$ equal to or higher than 0.95 is considered as a good fitting by linear regression. Therefore, according to the $r^{2}$ values generated by each model (Zero-order, First-order, Higuchi, Korsmeyer–Peppas and Weibull) for each membrane (CA, CAP, CAPS and CAPS), the models that best fit the release profiles were Korsmeyer–Peppas and Weibull. In the case of the Korsmeyer–Peppas model, $r^{2}$ values of all samples were higher than 0.95, but in the case of the Weibull model, the fitting for the CA and CAP membranes returned values of almost 0.95 and for the CAS and CAPS membranes, values were higher than 0.95.

Korsmeyer–Peppas is a model determined on the basis of experimental data and it is a shortened version of the solution of the diffusion equation by Crank (1975) [25]. In this model, the release mechanisms are dependent on the sample geometry (thin films, cylinders, spheres). In that sense, when the Korsmeyer–Peppas model is applied to thin films, such as the membranes of this study, if the release parameter *n* is less than 0.5, it means that drug diffusion is occurring through a partially swollen matrix and through a solution-filled network mesh [26]. If n=0.5, the Fickian diffusion is controlling the release and the solvent penetration is the rate-limiting step. On the other hand, if 0.5<n<1, then this is related to a non-Fickian release, which is the release of the extract controlled by both diffusion and erosion mechanisms. If n=1, then the release corresponds to the zero-order model, where the release of the extract is independent of time [9]. In that sense, according to the results from the Korsmeyer–Peppas model, the *n* values for the four types of membranes were lower than 0.5; this indicates that the release of the extract was controlled by a diffusion occurring through a partially swollen matrix, which could be produced by the polymer chains relaxation. Moreover, the $k_{\mathrm{KP}}$ parameter is related to the interaction between the extract and the constituent polymers of the membrane, and higher values mean higher rates of release. According to this, the $k_{\mathrm{KP}}$ values obtained for the samples presented the following tendency:

CAPS > CAP > CAS > CA.

This means that the porous membranes (CAPS and CAP) generated a faster release probably due to the pores of the polymeric matrix, which facilitated the absorption of water from the release medium (PBS). On the contrary, for membranes without pores (CAS and CA), which are more compact and dense, the absorption of water would be limited, and as a consequence, the release of the extract, too. As CAS and CA samples obtained higher amounts of extract release than CAPS and CAP (Figure 2), this means that faster release of CAPS and CAP was mainly controlled by the porous membrane cross-linked structure rather than the concentration differential. On the other hand, although the Weibull model has been criticized for the lack of a kinetic basis for its use and for the non-physical nature of its parameters, according to different works, the Weibull function demonstrated that the exponent β, for polymeric matrices, is an indicator of the mechanism of transport of the drug through the matrix related to the exponent *n* of the power law model [17,27,28,29]: a value of β less than or equal to 0.75 was associated with Fick diffusion in either fractal or Euclidean space, while a combined mechanism (Fick diffusion and swelling controlled transport) was associated with β values in the range 0.75 < β < 1. For β > 1, drug release involves complex mechanisms, which imply that the release rate does not change monotonically. In fact, the release rate initially increases nonlinearly up to an inflection point and then decreases asymptotically.

According to the above, the results obtained for the Weibull β parameter (Table 2) indicated that the mechanism of release was Fickian diffusion (β less than or equal to 0.75). However, it is also important to identify the polymeric matrix geometry in this β range [27]: for β < 0.35, diffusion occurs in highly disordered spaces, differently than the percolation cluster; for 0.35<β<0.69, diffusion occurs in a fractal substrate; for 0.69<β<0.75, diffusion takes place in a normal Euclidean space. These results suggest that the non-porous membranes (CA and CAS) present a fractal structure, and the porous formulations (CAP and CAPS) show highly disordered structures. Fractal organization of CA and CAS samples could be originated during the association of alginate carboxylic groups (-COO${}^{-}$) with chitosan amino groups (-NH3${}^{+}$) [7,30]; meanwhile, the polymeric matrices of CAP and CAPS were highly disorganized by the pores formation.

As a summary, the release of *A. chica* extract from all membrane formulations occurred by a Fickian difussion, what was confirmed by both Korsmeyer–Peppas and Weibull models. Moreover, the Weibull model suggested that non-porous membranes (CA and CAS) had fractal geometry and that porous membranes (CAP and CAPS) were highly disorganized structures.

Furthermore, in this work, the cost function is presented as a tool to analyze different mathematical models that simulate experimental data of release profiles of *A. chica* extract from C/A membranes to a greater extent. Accordingly, Figure 4 shows the cost function simulations applied to each model (Zero-order, First-order, Higuchi, Korsmeyer–Peppas and Weibull) in each membrane type (CA, CAS, CAP and CAPS), and Table 3 presents the parameters obtained. The lower the fitted cost function value (*F*), the more efficient the model. Conversely, when the cost function value is higher, the model is less effective.

According to the results obtained by the cost function, *F* values in Table 3, the best simulations were performed by

Weibull > Korsmeyer–Peppas > 
            > Higuchi > First-order > Zero-order,

which is similar to the previous results. However, it is important to highlight that using the cost function, the Weibull model fits better to the release profiles than the Korsmeyer-Peppas model for all the membranes. Note that in the case of the Weibull model $e_{1}$ and $e_{2}$ correspond to α and β, respectively, and the $e_{3}$ value, which is $T_{i}$, was not reported because it is zero for all cases (Table 2). According to the cost function results, CA, CAS and CAP formulations would present a Fickian diffusion through a polymeric matrix of fractal geometry, and only the CAPS membrane would have a highly disordered matrix. This partially different result compared to the case when the cost function is not applied can be due to the fact that *F* values of the Weibull model are around 10 times lower—i.e., 10 times mores precise—than the Korsmeyer–Peppas model. Without the use of the cost function, the $r^{2}$ of both models are very similar. In that sense, the cost function presents a significant advantage, which is a higher fitting sensitivity.

## 6. Conclusions

This work consisted on studying the performance of in vitro experimental *A. chica* extract release profiles obtained from four different formulations of chitosan/alginate membranes: a dense membrane (CA), a dense and flexible membrane (CAS), a porous membrane (CAP), and a porous and flexible membrane (CAPS). The mechanism of the extract release kinetics was determined comparing classic models from the literature: Zero Order, First Order, Higuchi, Korsmeyer–Peppas and Weibull. Furthermore, in order to improve the mathematical analysis between the models, and as the main novelty of this work, the cost function was presented as a tool to analyze these different mathematical models that simulated the release profiles and were compared to experimental data. Our method explored how some metrics and weights of the cost function impact on the results of the release models that describe experimental information for drug delivery release processes. Our results indicated that the use of the proposed model parameter optimization by the cost function, which better fits the experimental data, had the significant advantage of showing a higher fitting sensitivity.

## Figures and Tables

**Figure 1 polymers-14-01109-f001:**
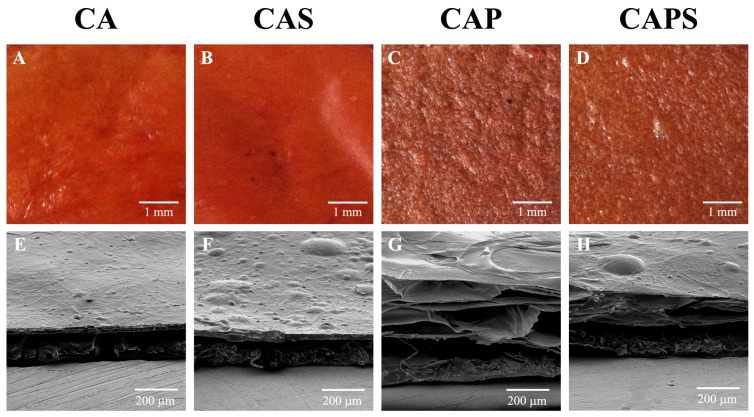
Photographs and SEM images of membrane samples loaded with *A. chica* extract: CA (**A**,**E**), CAS (**B**,**F**), CAP (**C**,**G**) and CAPS (**D**,**H**).

**Figure 2 polymers-14-01109-f002:**
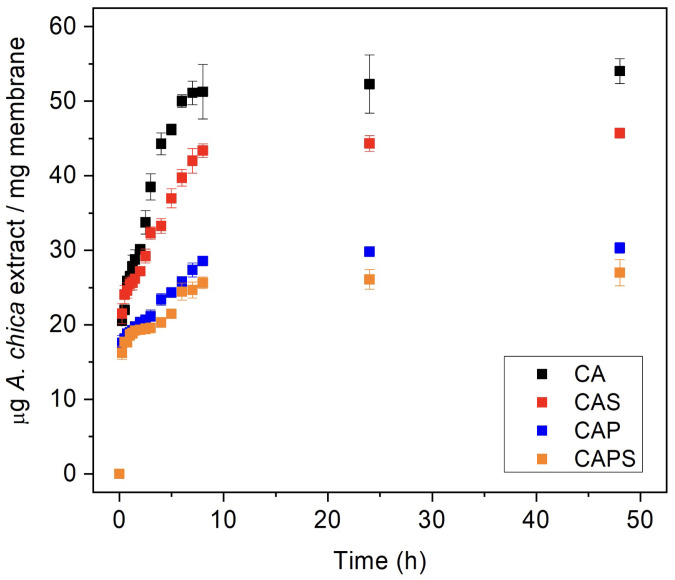
Experimental *A. chica* extract release profiles from the C/A membranes. Results represent mean ± SD of three measurements.

**Figure 3 polymers-14-01109-f003:**
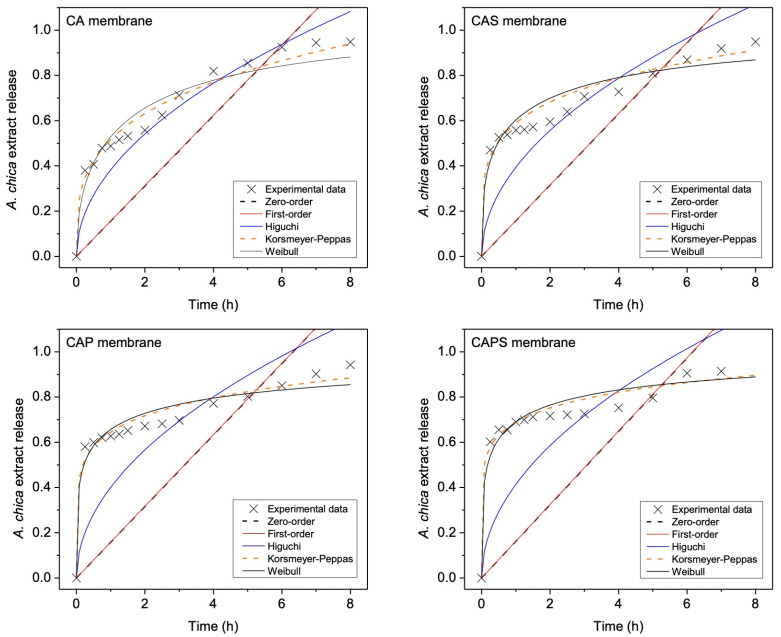
Simulations of the release profiles from each membrane (CA, CAS, CAP and CAPS) using the Zero-order, First-order, Higuchi and Korsmeyer–Peppas and Weibull models.

**Figure 4 polymers-14-01109-f004:**
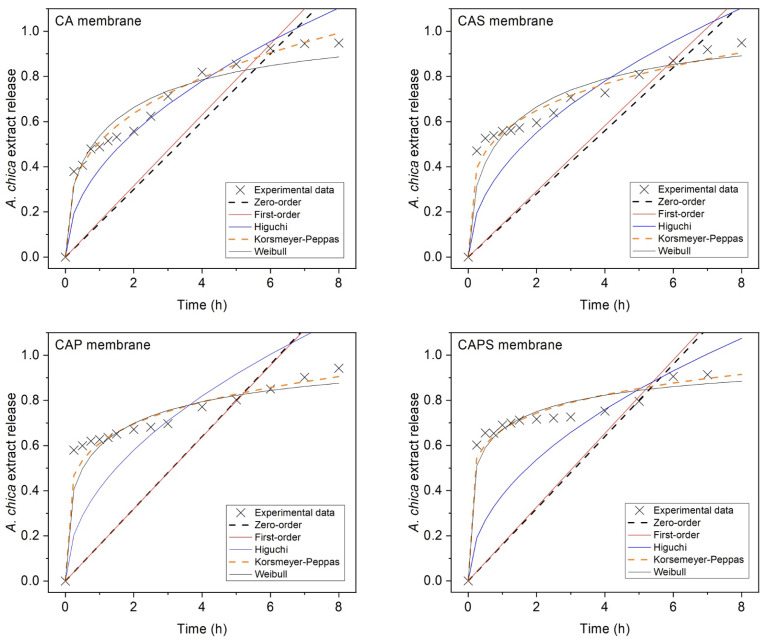
Cost function simulations of the mathematical models (Zero-order, First-order, Higuchi and Korsmeyer–Peppas and Weibull) used in each membrane: CA, CAS, CAP and CAPS.

**Table 1 polymers-14-01109-t001:** Model evaluation: Satisfaction status of criteria by considered models.

Model	Function	Q(0)=0	$\lim_{t\to \infty }Q\left(t\right)<\infty$
Zero-order equation	$Q\left(t\right)=Q_{0}+k_{0}t$	NO	NO
First order model	$Q\left(t\right)=\left(1-\exp\left(-k_{1}t\right)\right)$	YES	YES
Higuchi model	$Q\left(t\right)=k_{H}t^{1/2}$	YES	NO
Korsmeyer–Peppas model	$\frac{Q(t)}{Q_{\infty }}=k_{\mathrm{KP}}t^{n}$	YES	NO
The Weibull model	$Q\left(t\right)=1-\exp\left(-{\left(t-T_{i}\right)}^{\beta }/\alpha \right)$	YES	YES

**Table 2 polymers-14-01109-t002:** In vitro release kinetics of *A. chica* from C/A membranes.

Sample	Zero-Order		First-Order		Higuchi		Korsmeyer-Peppas			Weibull			
	$k_{0}$ (h${}^{-1}$)	$r^{2}$	$k_{1}$ (h${}^{-1}$)	$r^{2}$	$k_{H}$ (h${}^{-1/2}$)	$r^{2}$	$k_{\mathrm{KP}}$ (h${}^{-n}$)	*n*	$r^{2}$	α (-)	β (-)	$T_{i}$ (h)	$r^{2}$
CA	0.1556	−0.1982	0.0015	−0.1924	0.3827	0.7979	0.5170	0.2872	0.9818	1.7645	0.5014	0.0000	0.9476
CAS	0.1574	−0.9428	0.0015	−0.9351	0.3937	0.5193	0.5897	0.2095	0.9792	1.2335	0.3793	0.0000	0.9507
CAP	0.1580	−1.7429	0.0015	−1.7154	0.4005	0.1652	0.6455	0.1519	0.9720	1.2636	0.2843	0.0000	0.9487
CAPS	0.1602	−2.1602	0.0016	−2.1497	0.4139	−0.0494	0.6889	0.1264	0.9701	1.7013	0.3009	0.0000	0.9531

**Table 3 polymers-14-01109-t003:** Summary of results of the cost function fitting.

Results	CA	CAS	CAP	CAPS
Zero-order				
$e_{1}\left(k_{0}\right)$	0.153±0.006	0.143±0.005	0.162±0.008	0.159±0.004
*F*	89,011±6396	77,665±5433	99,925±9832	94,746±6789
First-order				
$e_{1}\left(k_{1}\right)$	0.340±0.013	0.360±0.003	0.512±0.007	0.561±0.009
*F*	15,192±3755	17,626±311	36,954±2754	31,566±1140
Higuchi				
$e_{1}\left(k_{H}\right)$	0.392±0.700	0.390±0.400	0.411±0.100	0.381±0.800
*F*	8076±302	7995±165	8901±44	7615±319
Korsmeyer-Peppas				
$e_{1}\left(k_{\mathrm{KP}}\right)$	0.514±0.004	0.549±0.010	0.61±0.008	0.671±0.003
$e_{2}\left(n\right)$	0.321±0.005	0.241±0.006	0.192±0.001	0.152±0.003
*F*	2388±102	2767±73	3570±97	4550±67
Weibull				
$e_{1}\left(\alpha \right)$	0.133±0.003	0.13±0.002	0.11±0.003	0.09±0.012
$e_{2}\left(\beta \right)$	0.521±0.005	0.51±0.003	0.402±0.005	0.32±0.006
*F*	292±2	271±6	298±5	316±5

## Data Availability

The data that support the findings of this study are available from the corresponding author upon reasonable request.

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
