# Peer review of "Cost Function Analysis Applied to Different Kinetic Release Models of Arrabidaea chica Verlot Extract from Chitosan/Alginate Membranes"

_polymers, 2022, doi:10.3390/polym14061109_

Round 1
Reviewer 1 Report
Dear authors,
Please find attached the review report.
All the best.

Reviewer 2 Report
The manuscript is innovative, but I have considerable doubts as to the applicability of the method proposed in the study. The models studied are semi-empirical models or are derived from purely mathematical functions (such as the Weibull function). It seems that no detailed analysis of the assumptions on which the tested models are based has been carried out. Modifications without such an analysis do not make sense from a physical point of view. The release process was investigated in this work, so I consider taking my comments into account to be helpful in assessing the value of the work presented.
Please refer to the following comments and clarify unclear issues:
1) The "Mathematical models" section shows the models used for the analysis. There is no source literature here. Please provide source articles for formulas 1-5, because the quoted articles [16, 17] are only review papers, not source articles. It is particularly important due to the conditions of applicability of the selected models (such as the geometry of the sample). Please look at publications such as: Peppas, N.A. Pharm. Acta Helv. 1985; Korsmeyer, R.W; Int. Jorunal Pharm. 1983; Weibull, W. J Appl Mech 1951; T. Higuchi, J Pharm Sci 1963.
2) Peppas model is a model determined on the basis of experimental data. It is actually a shortened version of the solution of the diffusion equation by J. Crank (The Mathematics of Diffusion, 1975). Note that the parameter limits (eg, "n") that determine the mechanisms are dependent on the sample geometry (disks, cylinders, rolls), see Ritger and Peppas, Journal of Controlled Release 5 (1) (1987) 23. I have not found information about the geometry of the tested sample.
3) It is particularly important to provide the source article for the version of the Weibull model used. The works cited, including those 21-23, are not source articles. As a result, there is no access to knowledge about the assumptions of the model application.
4) Swelling took place in a very small volume of solution. How long did the swelling process take? How can you be sure that there has been no reabsorption? Did the high concentration affect the absorbance readings? Therefore, I believe that the observed absorption spectra of the test substance should also be shown.
5) Why is t*= 8 h selected? According to Peppas (Pharm. Acta Helv. 1985), equation (4) can only be applied to the points corresponding to the initial mass change (up to about 60% of the total amount of mass transported).
6) If the models (solid lines) were matched to the data in Fig. 2, please provide them. If these are only experimental data, then the points should not be combined as this suggests a trend.
7) The conclusion obtained in the study is the Fickian diffusion mechanism responsible for the release process. This mechanism is characterized by a linear relationship between the amount of substance released and t^(1/2), where t is the release time (see T. Alfrey Journal of Polymer Science: Part C 12, (1966), 249). In line with the models described, such a relationship is presented in equation (3), but the authors conclude that the Pepass model best describes the process under study. There is no exact explanation.
8) What was the effect of the cross-linking of the tested materials on the release mechanism? Have relaxation processes been considered?
9) Dry samples were tested by means of SEM. Should they not be examined after they have been moistened and swollen? It seems that the change of the spatial structure also influences the processes taking place.
10) Swelling took place in a very small volume of the solution. How long did the swelling process take? How can you be sure that there has been no reabsorption? Did the high concentration affect the absorbance readings? Therefore, I believe that the observed absorption spectra of the test substance should also be shown.
11) Since the experiment was repeated three times, how were the data selected for analysis? Were mean values used? Please explain the analysis process.
Round 2
Reviewer 2 Report
Thank you for the well-described authors' replies. Again, congratulations on proposing an interesting modeling method to describe the release process.